

# West Antarctic sites for subglacial drilling to test for past ice-sheet collapse

Perry Spector[1], John Stone[1], David Pollard[2], Trevor Hillebrand[1], Cameron Lewis[3], and Joel Gombiner[1]

[1]Department of Earth and Space Sciences, University of Washington, Seattle, WA, USA
[2]Earth and Environmental Systems Institute, Pennsylvania State University, University Park, USA
[3]Center for Remote Sensing of Ice Sheets (CReSIS), University of Kansas, Lawrence, KS, USA

**Correspondence:** Perry Spector (spectorp@gmail.com)

**Abstract.** Mass loss from the West Antarctic Ice Sheet (WAIS) is increasing, and there is concern that an incipient large-scale deglaciation of the marine basins may already be underway. Measurements of cosmogenic nuclides in subglacial bedrock surfaces have the potential to establish whether and when the marine-based portions of the WAIS deglaciated in the past. However, because most of the bedrock revealed by ice-sheet collapse would remain below sea level, shielded from the cosmic-ray flux, drill sites for subglacial sampling must be located in areas where thinning of the residual ice sheet would expose presently subglacial bedrock surfaces. In this paper we discuss the criteria and considerations for choosing drill sites where subglacial samples will provide maximum information about WAIS extent during past interglacial periods. We evaluate candidate sites in West Antarctica and find that sites located adjacent to the large marine basins of West Antarctica will be most diagnostic of past ice-sheet collapse. There are important considerations for drill-site selection on the kilometer scale that can only be assessed by field reconnaissance. As a case study of these considerations, we describe reconnaissance at sites in West Antarctica, focusing on the Pirrit Hills, where in the summer of 2016-2017, an 8 m bedrock core was retrieved from below 150 m of ice.

## 1   Introduction

There is strong but indirect evidence for a diminished West Antarctic Ice Sheet (WAIS) during some past warm interglacial periods of the Pleistocene. Coastal records of former sea-level highstands (see reviews in Dutton et al., 2015), along with geological evidence from Antarctica (Scherer et al., 1998), ice-sheet modeling experiments [e.g. Pollard and DeConto, 2009], and other lines of evidence [e.g. Barnes and Hillenbrand, 2010; NEEM community members, 2013; see review in Alley et al., 2015] suggest large-scale deglaciation of the WAIS within the past ~1 Myr, which may have occurred as recently as the last interglacial period, ~125 kyr ago. It has also been suggested that the WAIS, along with the Greenland Ice Sheet and parts of the East Antarctic Ice Sheet, disappeared during the mid-Pliocene (~3 Myr ago), the last time atmospheric $CO_2$ concentrations reached modern levels. However, the evidence for this, which largely consists of relative sea-level data and marine oxygen isotope records, has large uncertainties which preclude robust estimates of sea-level and ice-sheet volume during this period [see reviews in Dutton et al., 2015 and Raymo et al., 2017].

Large-scale deglaciation of the WAIS (so-called "collapse") is theorized to occur because much of the ice sheet overlies deep marine basins in a configuration that makes it susceptible to a feedback between marginal retreat, flow acceleration,





thinning, and flotation (Weertman, 1974; Schoof, 2007). Increasing ice loss is presently occurring through these mechanisms
in the Amundsen Sea sector of the WAIS (Fig. 1) (see review in Scambos et al., 2017), and numerical modeling suggests that
an incipient collapse of the ice sheet is underway in this sector (e.g. Joughin et al., 2014). How much and how quickly future
sea level will rise due to WAIS deglaciation remains unknown (Scambos et al., 2017), but continued ice loss could eventually
increase global mean sea level by 3-4 m (Bamber et al., 2009). Knowledge of ice-sheet extent during interglacials warmer and
more prolonged than the Holocene would be invaluable for understanding, and potentially predicting, the future stability or
instability of the WAIS. Geological observations from West Antarctica which constrain the configuration of the WAIS during
former interglacial periods remain scarce because evidence of its limits during these times is concealed beneath the present-day
ice sheet.

One potential source of evidence is the presence or absence of long-lived cosmogenic nuclides in subglacial bedrock. Be-
cause most cosmic radiation is absorbed by as little as 5-10 m of ice cover, discovering significant concentrations of these
nuclides would provide unambiguous evidence for former ice-free conditions and could establish whether the WAIS collapsed
during the past few million years. At sites affected by a single collapse, cosmogenic-nuclide measurements can directly date
that event. In the case of more complex deglaciation histories, the same data record the cumulative exposure time of the bedrock
surface. Although cosmogenic nuclides have the potential to unambiguously indicate past ice-sheet collapse on timescales rang-
ing from the Holocene to the Pliocene, the power of the method depends on careful site selection. In this paper we describe the
criteria and considerations for choosing subglacial sampling sites where cosmogenic-nuclide data will provide the maximum
amount of information about WAIS extent during past interglacials.

In central Greenland, a bedrock core was opportunistically retrieved from below the full thickness of the ice sheet at the
GISP2 drilling site. Concentrations of cosmogenic $^{10}$Be and $^{26}$Al in the core require periods of prolonged exposure during the
Pleistocene when the ice sheet was largely absent (Schaefer et al., 2016; Nishiizumi, 1996). In contrast to this work, bedrock
recovered from below the thick portions of the WAIS would not be capable of providing equivalent information because the
bed in these areas is located far below sea level (Fig. 1) and would remain submerged, shielded from the cosmic-ray flux, if the
ice sheet collapsed. Establishing whether the thick, marine-based portions of the WAIS disappeared in the past will therefore
require drilling through adjacent, thinner portions of the ice sheet into subglacial highlands that would be exposed by ice
thinning during collapse events.

In the 2016-2017 summer, the first cores of subglacial bedrock from West Antarctica were recovered from the Pirrit Hills
and the Ohio Range (Fig. 1), which was made possible by recent advances in sub-ice drilling technology (e.g. Goodge and
Severinghaus, 2016). For these as well as future drilling efforts to provide meaningful information about past WAIS config-
urations, the measurements made on the recovered bedrock must be representative of the past ice-thickness at the drill site,
which in turn must be linked to the extent of the broader ice sheet. In Section 2 of this paper, we describe cosmogenic-nuclide
considerations that guide drill-site selection. In Section 3, we use an ice-sheet model to predict the areas of the WAIS where
significant thinning (and thus exposure of presently subglacial bedrock) would occur during collapse events. In Section 4, we
evaluate a group of candidate drill sites throughout West Antarctica. Finally, in Section 5, we describe reconnaissance work at



three sites in West Antarctica, with emphasis on the Pirrit Hills, which we present as a case study of drill-site selection on the scale of an individual nunatak.

## 2 Cosmogenic nuclide considerations for drill site selection

### 2.1 Strategies for subglacial bedrock sampling and analysis

Because subglacial drilling is expensive and time consuming, drill-site selection, drilling operations, and analysis of recovered samples should be designed to maximize the information provided by the inherently limited amount of subglacial bedrock. There are several strategies to accomplish this. At a given drill site, collecting multiple bedrock cores in an elevation transect below the modern ice surface can establish the magnitude of past deglaciations. By locating drill sites near outcropping mountains, elevation transects can be extended up to and above the limits of the thicker, ice-age WAIS, thereby constraining ice-thickness variations over the full glacial-interglacial cycle. Drilling near outcrops also allows the subglacial rock type to be inferred with confidence, which is important as not all lithologies are suitable for cosmogenic-nuclide measurements. Although measuring a single cosmogenic nuclide (e.g. $^{10}$Be or $^{36}$Cl) in subglacial bedrock samples is enough to detect past exposure, measuring several nuclides that have different half lives yields considerably more information about the glacial history, both in the recent and the distant past. Independent constraints on the most recent period of exposure can potentially be added by (i) collecting and dating the basal ice and (ii) luminescence dating of the subglacial bedrock surface. Because cosmogenic nuclides are primarily produced within the topmost few meters of the bedrock surface, preservation of the record of past exposure requires drill sites to be located in areas where erosion below or above the ice has been minimal. Knowledge of the erosion history, which is required for accurate interpretation of the glacial history, can be gained by analyzing not only surface samples of the subglacial bedrock, but by measuring depth profiles in rock cores that extend several meters below the surface.

### 2.2 Subglacial bedrock lithology

The central portion of West Antarctica is composed of three tectonic and topographic blocks, the Ellsworth-Whitmore Mountains, Marie Byrd Land, and what is known as the Thurston Island region, which are separated by low-lying areas of the West Antarctic Rift System (Fig. 1). The portions of the bed that would be above sea level if the WAIS collapsed are primarily located in these three tectonic regions; however there are isolated peaks and plateaus, such as subglacial Mt. Resnik (Behrendt et al., 2007), which rise above sea level from the deep marine basins (Fig. 1).

Drilling into these subglacial highlands must target rock types in which useful cosmogenic nuclides can be measured. Commonly measured nuclides (and their half-lives) include $^{10}$Be (1.4 Myr), $^{26}$Al (0.7 Myr), $^{21}$Ne (stable), $^{36}$Cl (0.3 Myr), $^{3}$He (stable), and $^{14}$C (5.7 kyr). If more of these nuclides can be measured, more restrictive time constraints can be placed on exposure episodes that may have occurred in the distant and/or recent past. Because (i) all of these nuclides (except $^{36}$Cl) can be measured in quartz, and (ii) their production rates are best known for quartz, quartz-rich rocks would allow the glacial history to be constrained over a large range of timescales. Igneous or metamorphic rocks of granitic composition would permit the

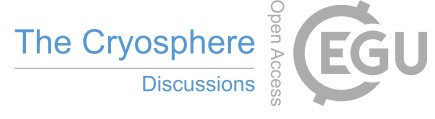

greatest variety of analyses because, in addition to containing quartz, they also contain potassium feldspar and (commonly) Cl-rich mica in which $^{36}$Cl can be measured. Other analytical strategies could be applied to the volcanic rocks that underlie large

areas of the WAIS. For example, cosmogenic $^{3}$He, $^{21}$Ne, and $^{36}$Cl could be measured in basaltic pyroxene, or in a combination of quartz and sanidine from more felsic volcanic rocks. However, the relative abundance of useful minerals such as these is also important because of the small amount of rock that can be retrieved by subglacial drilling (Goodge and Severinghaus, 2016).

The necessity of recovering a suitable rock type implies that it will likely be advantageous to target the subglacial extension of outcropping mountains, where the rock type at depth can be inferred with confidence. Although geophysical surveys can

narrow the range of possible lithologies, the precise identity and mineralogy of underlying bedrock generally remains unknown (e.g., Behrendt et al., 1994; Jordan et al., 2013). In this paper, we largely restrict our consideration of potential drill sites to areas near mountains of granitic composition. Figure 1 shows the location of some of the granitic nunataks in West Antarctica. Although this map is not comprehensive, it is generally representative of their geographic distribution. Scattered granitic nunataks outcrop between the Ellsworth Mountains and the southern Transantarctic Mountains, predominately in the Weddell

Sea Sector of the WAIS. Granitic nunataks are also located (i) at the base of the Antarctic Peninsula, (ii) in the Thurston Island region, and (iii) along sections of the Marie Byrd Land coast. In addition to granitic sites, in Fig. 1 we also include quartzite peaks of the northern Ellsworth Mountains; the isolated quartzite nunatak of Mt. Johns; and subglacial Mt. Resnik, which, although likely volcanic (Behrendt et al., 2007), is a tempting drill target because (i) it lies upstream of where Scherer et al. (1998) found evidence for a large-scale Pleistocene deglaciation of the WAIS; (ii) its conical form and high relief suggest that

it erupted subaerially when the ice sheet was absent (Behrendt et al., 2007); and (iii) its summit is only ∼330 m below the ice surface (Morse et al., 2002).

## 2.3 Preservation of the cosmogenic-nuclide record

Because the cosmogenic-nuclide record is primarily produced in the topmost few meters of an exposed bedrock surface, its survival requires that the bedrock remain continuously protected from erosion. This is most likely to be the case in areas

that are surrounded by slow-flowing ice and where thickening of the ice sheet during past glacial periods such as the LGM was minimal. Other factors which promote cold-based ice are high accumulation rates, low surface temperatures, and a low geothermal heat flux. As discussed in Section 4, sites in the WAIS interior are likely to have preserved subglacial bedrock surfaces hundreds of meters below the modern ice level. Preserved subglacial surfaces also likely exist near the ice-sheet margin, however, it may be more difficult to identify these sites with confidence. Although subglacial bedrock samples that

have remained continuously uneroded will provide the greatest constraints on the glacial history, samples that have experienced low rates of erosion may also be of use, provided that the erosion history can be estimated. As mentioned above, this can be done by measuring cosmogenic nuclides not only in the subglacial bedrock surface, but also in depth profiles along the length of short (∼3-6 m) bedrock cores (Ploskey and Stone, 2012).

A related concern to bedrock erosion is the possibility that presently subglacial surfaces remained concealed by till, soil, or snow when the ice sheet disappeared in the past. Failure to account for past surface cover would cause the true exposure history to be underestimated. Analysis of the subglacial bedrock core from the GISP2 site in central Greenland suggests that





the present-day bedrock surface was covered by a thin layer of material when the ice sheet disappeared during the Pleistocene (Schaefer et al., 2016). Soil accumulation there is plausible because debris-rich basal ice in the GISP2 and other Greenland ice

cores, contain evidence for a vegetated landscape during one or more interglacial periods in the past million years (Willerslev et al., 2007; Bierman et al., 2014; Souchez et al., 2006). In Antarctica, however, fossil organisms and pollen from sites spanning the continent show that a tundra landscape went extinct around the mid-Miocene, and that the climate has been continuously polar since that time (Lewis et al., 2008; Ashworth and Erwin, 2016; Anderson et al., 2011; Wei et al., 2014). Although soil is unlikely to have covered drill targets in Antarctica during former interglacial periods, accumulated till is possible. In the

vicinity of outcropping mountains where drill sites will probably be located, englacial debris commonly accumulates in blue-ice areas, and subsequent thinning could drape the underlying bedrock with a layer of till. As discussed below in Section 5, this concern can be mitigated by locating drill sites above subglacial ridges where the likelihood of till or snow cover is minimal.

## 3   Where will WAIS collapse cause the largest changes?

WAIS collapse is theorized to occur via a feedback in which initial retreat of the grounding line into deeper water causes

more ice to flow across grounding line (Weertman, 1974; Schoof, 2007). This accelerates the flow of the ice sheet upstream, causing it to thin, which in turn causes previously grounded ice to float as the grounding line recedes farther inland. The thinning of upstream grounded ice is important because it is the link between withdrawal of ice from the marine basins and the exposure of presently subglacial bedrock surfaces in the WAIS interior. Therefore, an overarching criterion is that drill sites be located in areas that experience the largest change in ice thickness during collapse events. Recent observations in the

Amundsen Sea sector show that thinning induced by grounding-line retreat is greatest near the ice-sheet margin, but remains detectable hundreds of kilometers upstream (Pritchard et al., 2012). This suggests that although large portions of the WAIS are prone to thinning during deglaciations, some sites will be more or less diagnostic of past ice-sheet collapse.

The geological and glaciological constraints on WAIS configuration during times of reduced ice volume are scant (Scherer et al., 1998; Korotkikh et al., 2011; Mulvaney et al., 2014) and therefore of limited use for assessing the response of candidate

drill sites during collapse events. A much more complete record exists for the deglaciation of marine-based ice in the Ross Sea following the LGM, which may provide a useful analog. Withdrawal of ice from the outer Ross Sea, where the ice-sheet was relatively thin, led to moderate thinning of ∼300 m in adjacent areas of northern Victoria Land [Goehring, pers. comm., 2018]. Moderate thinning also occurred at Siple Dome (Waddington et al., 2005; Price et al., 2007), which overlies a broad area of high topography, as well as at the Ohio Range (Ackert et al., 2007), which remains well upstream of the modern grounding line.

The greatest thinning (>1 km), however, is recorded at sites in the southern Transantarctic Mountains that directly abut a deep marine basin in the western Ross Sea which lost ∼1.5-2 km of ice (Spector et al., 2017; Bromley et al., 2012). Collectively, these observations suggest that if the WAIS collapsed during past interglacial periods, the greatest thinning likely occurred at sites directly adjacent to the deep marine basins in West Antarctica that became ice free, such as Bentley Subglacial Trench (Fig. 1).





In order to examine the transient response of different parts of the WAIS to collapse events, and thereby identify which areas will be most diagnostic of past deglaciations, we use the Penn State University ice-sheet model (PSU-3D) to simulate the Antarctic Ice Sheet continuously over the past 5 Myr. This time period, from the early Pliocene to the present, covers a large range of glacial-interglacial climates and is comparable to the time period that can be investigated with cosmogenic-nuclide measurements. In comparison, most existing collapse simulations depict only a single deglaciation [e.g. Feldmann and Levermann, 2015] or are equilibrium models [e.g. de Boer et al., 2015] which may not represent ice-sheet behavior during short-lived events such as Pleistocene interglacial periods.

The model we use solves a hybrid combination of the scaled dynamical equations for the flow of grounded and floating ice (Pollard and DeConto, 2009, 2012a, b). It is similar to that presented in Pollard and DeConto (2009), however it uses basal sliding coefficients derived from inverse calculations, as well as improved parameterizations of ice-shelf calving and sub-ice oceanic melting (Pollard and DeConto, 2012b). A parameterization by Schoof (2007) allows for reasonably accurate simulations of grounding-line migration on the coarse grids that are required for multi-million year model runs. We use a 40 km horizontal grid, which, although coarse, is comparable to the resolution with which the bed is known in the vicinity of many of the candidate drill sites (Fretwell et al., 2013). Bedrock deformation in the model is treated as an elastic lithosphere above a viscous asthenosphere that relaxes toward isostatic equilibrium. The model includes two processes that exacerbate retreat during warm climates: (i) hydrofracture of ice shelves due to surface water draining into crevasses and (ii) structural failure of ice cliffs at the grounding line (Pollard et al., 2015).

The model is forced with parameterizations of surface temperature, precipitation, sub-ice-shelf melting, and sea level, which have been described in previous publications (Pollard and DeConto, 2009, 2012a). These parameterizations are largely functions of a stacked benthic $\delta^{18}O$ record (Lisiecki and Raymo, 2005) and orbital insolation variations (Laskar et al., 2004). In this run, we add the influence of long-term atmospheric $CO_2$ decline, by prescribing a linear ramp from 400 ppmv to 280 ppmv $CO_2$ between 3 Ma and 2 Ma, with corresponding small uniform shifts to atmospheric and oceanic temperatures. This results in generally smaller model ice volumes prior to 3 Myr compared to Pollard and DeConto (2009). The parameters for the simulation shown in Fig. 2 have been calibrated in previous model experiments [e.g. Pollard and DeConto, 2012a, b; Pollard et al., 2016]. Because many of the parameters related to the climate forcing, as well as other aspects of the model physics, are uncertain, and alternative values can affect the size of the ice sheet during interglacial and glacial periods as well as its rate of change, we use this simulation as a guide to how ice thickness at candidate drill sites responds to deglaciation of the marine basins rather than an accurate depiction of the ice sheet through time.

The model is sampled every 5000 years, which sufficiently captures the ice-sheet's variations and results in minimal aliasing. The modeled ice sheet transitions rapidly between expanded and contracted configurations on orbital frequencies that reflect the climate forcing (Fig. 2). The simulated ice sheet was smaller than present prior to ~2.7 Myr ago, at which time its average size began to increase, and collapsed configurations with little or no marine-based ice in West Antarctica became less frequent. Some warm periods of the Pleistocene resulted in full deglaciation of the marine basins, leaving small ice sheets on areas of high topography, as shown in Fig. 2b. During other interglacial periods, thinning and grounding-line retreat were more limited and seaways were unable to link the Amundsen, Ross, and Weddell Seas. Although accumulation rates increase over



the residual ice sheet during deglaciations, in most areas this is insufficient to offset the dynamic thinning. Thinning is typically greatest directly upstream of the retreating grounding line, especially in areas adjoining deep marine basins that become ice free, as expected from the considerations described above.

## 4 Evaluation of candidate drill sites

In this section we use the ice-sheet model described above and available geologic information to evaluate candidate drill sites in terms of (i) how changes in local ice levels are related to the extent and configuration of the broader ice sheet, and (ii) whether the cosmogenic-nuclide record of past exposure is likely to have remained preserved by cold-based ice cover.

### 4.1 Sensitivity of sites to deglaciation of different parts of the WAIS

During collapse episodes simulated by the ice-sheet model, the grounding line retreats rapidly over deep marine basins, and is ultimately halted once it reaches shallow water. Because further retreat can only occur slowly and in the presence of a warm atmosphere, the ice-sheet margin is commonly pinned to a narrow band that closely follows the perimeter of the highland regions (Figs. 1 and 2). This is not unique to our simulation; rather it is a robust feature of ice-sheet models forced with interglacial climates [e.g. Bamber et al., 2009; DeConto and Pollard, 2016; Feldmann and Levermann, 2015; de Boer et al., 2015; Golledge et al., 2017]. The model demonstrates that in areas that remain glaciated, thinning occurs as the grounding line approaches, but then stops once the grounding line stabilizes, even if deglaciation continues in other sectors of the WAIS. The implication of this is that the magnitude of thinning at candidate drill sites is most directly controlled by the proximity of the grounding line and the thickness of ice lost from the marine basins immediately downstream, and less by ice-sheet changes elsewhere in West Antarctica.

At sites not adjoined to large marine basins, such as the Ford Ranges and the Jones Mountains, it may be difficult to know whether subglacial evidence of past exposure resulted from large-scale WAIS deglaciation or from modest retreat of the ice-sheet margin locally. In a related way, ice thinning at the Pirrit Hills, Nash Hills, and Pagano Nunatak is likely to be controlled most strongly by deglaciation of Robin Subglacial Basin, which underlies Institute and Möller Ice Streams (Fig. 1). Therefore, these sites may not be as sensitive to the presence or absence of grounded ice in the much larger basins of the Amundsen and Ross Sea sectors of the WAIS. A potential caveat to this is that our simulation, as well as others using the PSU-3D ice-sheet model [e.g. DeConto and Pollard, 2016], suggests that the large marine basins deglaciate in unison, implying that evidence for deglaciation from one sector could be extrapolated to other areas in West Antarctica. At sites on the periphery of the large basins in the Ross and Amundsen Sectors (i.e. Mt. Murphy, Mt. Resnik, the Ohio Range, the Whitmore Mountains, Mt. Woollard, Mt. Johns, and the northern Ellsworth Mountains), ice levels will likely be diagnostic of whether full collapse of the WAIS has occurred in the past.





## 4.2 Magnitude, timing, and frequency of thinning

The areas where the model predicts large drawdowns, not only during the most severe interglacial periods (which occur during the Pliocene in the simulation) but also during briefer warm periods of the Pleistocene, are the most likely areas to find evidence of past deglaciation in the form of previously exposed subglacial bedrock. The two sites that exhibit the greatest and most consistent ice loss are Mt. Woollard and Mt. Johns, which thin by up to ∼800 m during the Pliocene, the early

Pleistocene, and the late Pleistocene (Fig. 2e, i). The frequency of deglaciation at these sites is perhaps not surprising as they are located at the head of the Thwaites Glacier catchment, where (i) there is concern at present about the stability of the grounding-line [e.g. Scambos et al., 2017], and (ii) there are no major topographic obstacles to impede the grounding line until it reaches the Ellsworth-Whitmore Mountains. The large magnitude of deglaciation at these sites results from their proximity to Bentley Subglacial Trench, where more than 3 km of ice would be lost during large deglaciations. Most of the other sites in

the Ellsworth-Whitmore Mountains are predicted to thin consistently and significantly during collapse episodes; however, the greatest thinning is commonly restricted to the prolonged warm climates of the Pliocene, and does not occur, or does so only rarely, during the briefer Pleistocene interglacial periods (Fig. 2).

    Almost no thinning is predicted at Mt. Petras or the Thiel Mountains, which are not located upstream of deep marine basins that are vulnerable to deglaciation, and thus are not expected to experience dynamic thinning as a result of WAIS collapses. At

Mt. Moulton, a peak ∼175 km from Mt. Petras, ice has been found in an ablation area which dates to the last interglacial period, and as far back as ∼500 kyr B.P. (Korotkikh et al., 2011; Dunbar et al., 2008; Wilch et al., 1999). Although the presence of this ice does not require continuous glaciation of other portions of Marie Byrd Land, it is consistent with the simulation of stable ice levels shown in Fig. 2q. Minimal thinning is also predicted at Haag Nunataks, which is surprising because, in contrast, this site is surrounded by marine basins that deglaciate during collapse episodes. The site is unique, however, in that during

simulated collapse events it is located within a small ice cap that is represented by as few as 6 grid cells (Fig. 2b). Unlike the majority of the WAIS, these cells exhibit high spatial variability, with adjacent cells thinning and thickening, respectively, at times (which cancel out to produce the modest response shown in Fig. 2h). This example serves as a cautionary reminder that model results should not be over-interpreted, especially in areas where inadequate grid resolution results in spatial patterns of thinning and thickening that do not vary smoothly.

As shown in Fig. 2c, one possible complication predicted by the model is modest thinning in the WAIS interior during glacial periods (Steig et al., 2001). This is potentially important at the Whitmore Mountains, Mt. Woollard, and Mt. Johns because bedrock surfaces, covered by less than ∼100 m of ice, may have been exposed not only during collapse events, but also when the ice sheet was larger. Although Mt. Resnik is also located in the region expected to thin, its summit is ∼330 m below the surface (Morse et al., 2002) and would likely remain fully ice-covered. The thinning is caused by reduced accumulation over

the ice-sheet interior due to the decreased ability of the cold glacial atmosphere to carry moisture. Although similar thinning has been simulated previously [e.g. Golledge et al., 2012], most models of the LGM, including ones using the PSU-3D ice-sheet model, depict thicker-than-present ice (e.g. Briggs et al., 2014). The only existing geologic constraints come from exposure dating at Mt. Waesche and the Ohio range, sites on the margin of the region predicted to thin. These data indicate that ice was





modestly thicker at ∼10 kyr B.P. (Ackert et al., 1999, 2007); however, because this is several thousand years after accumulation rates began to rise in West Antarctica (Fudge et al., 2016), the data do not preclude thinner ice prior to ∼10 kyr B.P. Although lower ice levels during glacial periods could complicate the search for evidence of past ice-sheet collapse, determining whether ice levels in the WAIS interior were, in fact, lower during the LGM would be significant in its own right. Because thinning in the interior is expected to be less than ∼200 m (Fig. 2c), it would likely be possible to drill to deeper depths to encounter
bedrock that may have only been exposed during collapse episodes.

### 4.3    Preservation of subglacial bedrock surfaces

The ice-sheet model could, in theory, be used to predict erosion at candidate drill sites; however, the 40 km grid resolution is too coarse for this purpose, given that drill sites will likely be located near nunataks where topographic relief is high. Even if the model was run at higher resolution, insufficient knowledge of the geothermal heat flux and other factors that influence
basal conditions would preclude accurate erosion predictions. Instead we use the model as a guide to the relationship between three factors: ice thickness, surface velocity, and basal velocity (Fig. 3). Thickness and surface velocity are known or easily measured, while basal velocity, a strong indicator of erosion or preservation, is generally unknown or difficult to infer from measurements. Figure 3 shows that at the depths of interest for subglacial drilling in West Antarctica (up to ∼1 km), significant glacial erosion is unlikely in areas where the surface velocity is less than ∼10 m yr$^{-1}$. Such areas are common in the ice-sheet
interior, as well as in some areas near the margin (Fig. 1). This result is consistent with other ice-sheet model experiments investigating the distribution of subglacial erosion in Antarctica (Jamieson et al., 2010).

Although slow flowing ice is present near many of the candidate drill sites shown in Fig. 1, the lowest velocities occur at the ice divides in the interior, near sites such as the Whitmore Mountains and Mt. Woollard. At these interior sites, where (i) surface temperatures are very low, and (ii) ice either thickened modestly during glacial periods [Ackert et al., 2007, 1999; Section 5]
or potentially thinned in some areas (Fig. 2), subglacial drill targets, to depths of at least a few hundred meters, have likely remained continuously frozen. This is supported by field observations and exposure dating at the Pirrit Hills, Ohio Range, Nash Hills, and Whitmore Mountains [Mukhopadhyay et al., 2012; Ackert et al., 2007; Section 5] which show that bedrock surfaces near the modern ice level are commonly weathered and have exposure ages of hundreds of thousands of years, implying that preserved bedrock surfaces likely extend below modern ice levels.

Sites near the modern ice-sheet margin, such as Haag Nunataks, Mt. Murphy, and the Jones Mountains, are predicted to thicken by up to ∼800-1000 m during glacial periods (Fig. 2), which suggests that bedrock surfaces there may be vulnerable to erosion beneath thick ice that is sliding at its base. Geologic constraints on former highstands are not available at these sites; however field observations from a site near Mt. Murphy indicate that ice was at least ∼300 m thicker than present during the LGM (Johnson et al., 2008). Of all the candidate sites that are located near the coast, the most extensive investigation of former
ice cover has been conducted at the Ford Ranges (Stone et al., 2003; Sugden et al., 2005), a group of peaks which extend ∼100 km inland from the modern grounding line (Fig. 1) and span multiple grid cells in the ice-sheet model. The model predicts moderate thickening during glacial periods at the upstream edge of the Ford Ranges (Fig. 2s), but considerably more near the modern grounding line (up to ∼900 m; not shown in Fig. 2), which is consistent with geologic constraints on LGM ice levels



(Stone et al., 2003). Sugden et al. (2005) reported evidence of wet-based glacial erosion on the lower flanks of many of the peaks in the Ford Ranges, especially those closest to the modern grounding line. Therefore, if uneroded subglacial bedrock surfaces exist here, they will most likely be found at shallow depths near the inland peaks, where past thickening was more limited and ice velocities are lower.

In contrast to the evidence for glacial erosion, it should be noted that other sites near the ice-sheet margin, such as the Pensacola Mountains (Fig. 1), show evidence for surface preservation by cold-based ice cover (Balco et al., 2016; Bentley et al., 2017). Additionally, radar data over ice rises around the perimeter of Antarctica indicate that the majority of them are currently frozen to their beds (Matsuoka et al., 2015). Although this was not necessarily the case during past glacial periods, the radar data suggest that cold-based ice may, in fact, be common near slow-flowing areas of the ice-sheet margin. Taken together, these observations indicate that although preserved subglacial bedrock surfaces likely exist at some coastal sites, it may be challenging to predict their locations with confidence.

## 4.4 Bedrock lithology

With the possible exception of Mt. Resnik, which is fully ice covered, the sites shown in Figs. 1 and 2 all have quartz-bearing bedrock that would allow for measurements of a wide range of cosmogenic nuclides. However, at some sites, the exposed rock also includes lithologies in which the ability to make cosmogenic-nuclide measurements would be limited, or, in some cases, impossible. These sites are the Nash Hills (see Section 5), Mt. Petras (Spiegel et al., 2016), the Jones Mountains (Rutford and McIntosh, 2007), and Mt. Johns (Storey and Dalziel, 1987). Drilling at these sites may therefore remain risky unless the subglacial distribution of rock types can be determined.

## 5 Drill site reconnaissance

As discussed in Section 2 there are advantages to drilling in the neighborhood of exposed nunataks. However, the effects of nunataks on the local ice flow and meteorology cannot be captured at the scale of the ice-sheet model used in Sections 3 and 4, introducing considerations that need to be addressed by field reconnaissance. Fieldwork prior to drilling also allows for (i) sampling of exposed bedrock to test its antiquity with cosmogenic-nuclide measurements, (ii) determining the limits of glacial-to-present ice-sheet fluctuation, (iii) examining weathering features for evidence of rock surface preservation, and (iv) locating potential drill sites using ice-penetrating radar.

In 2012-13 we visited three sites in West Antarctica - the Pirrit Hills, Nash Hills, and Mt. Seelig in the Whitmore Mountains - to evaluate their potential for subglacial drilling. These three sites were selected as lying as closely as possible to a single flowline, with the (perhaps optimistic) idea of subglacial drilling along such a transect. In the end, most reconnaissance work was carried out in the Pirrit Hills; time constraints and problems with radar equipment limited exploration at the other two sites. Nonetheless: (i) Brief reconnaissance at the Nash Hills revealed rock types not suitable for cosmogenic-nuclide measurements and complicated bedrock structure that will require geophysical surveying to locate drill sites above subglacial granite. (ii) At the Whitmore Mountains, long-lived cosmogenic-nuclide measurements (which will be described in a forthcoming publication)





show evidence of prolonged exposure and limited ice-thickness changes. This, combined with the implication from the model that modest thinning may occur here during glacial periods (see Fig. 2, Section 4), discouraged us from selecting this site for initial subglacial drilling. (iii) Like the Nash Hills, the Pirrit Hills rise from the ice sheet approximately half way from divide to grounding line, where regional ice flow velocities are less than 5 m yr$^{-1}$ (Rignot et al., 2011). Here we were able to obtain evidence of low bedrock erosion, sizable glacial-interglacial fluctuations in ice cover, and radar profiles that revealed potential

drill sites. Based on these and other factors discussed below, we chose to drill at two sites near Harter Nunatak, a minor outcrop ∼5 km north of the Pirrit Hills massif.

## 5.1    Evidence for exposure and preservation of subglacial bedrock surfaces

At the Pirrit Hills, minimally weathered glacial deposits were found up to ∼330 m above the modern ice surface, marking the LGM highstand. This is similar to the thickening predicted by the ice-sheet model (Fig. 2f). Despite this evidence for thicker

ice, there is almost no indication of recent glacial erosion. Bedrock surfaces are oxidized, and, in places, exhibit case hardening, wind polish, and cavernous weathering pits (Fig. 5). Weathered surfaces occur both high on mountain flanks as well as near the modern ice surface, where they commonly intersect and appear to descend below the ice. Similar evidence for lower ice levels in the past has been found in other parts of East and West Antarctica (Mercer, 1968; Lilly et al., 2010; Mukhopadhyay et al., 2012). Although these observations demonstrate that ice levels were lower in the past; they neither establish the magnitude nor

the timing of thinning.

We collected two samples of weathered bedrock from Harter Nunatak (Figs. 4, 6) for analysis of cosmogenic [10]Be and [26]Al to determine the exposure, ice-cover, and erosional history of the nunatak. Analytical methods are described in the supplementary information. Cosmogenic nuclide data for other samples collected from the Pirrit Hills will be described in a subsequent publication. Data from these samples require minimum cumulative exposure of ∼630-650 kyr and minimum

cumulative ice cover of ∼350-460 kyr (Fig. 6). Together with the geomorphic observations, this indicates that (i) the ice at the Pirrit Hills has been both thinner and thicker than present for prolonged periods in the past ∼1 Myr, and (ii) during this time, bedrock surfaces have remained preserved by the polar climate and cold-based ice cover.

Firn temperatures in the vicinity of the Pirrit Hills are approximately -26°C and the ice is undoubtedly frozen to bedrock within a few hundred meters of the surface. Increasing ice thickness by ∼330 m during the LGM is unlikely to have raised

basal temperatures to near-melting. As discussed below, ice surface velocities at the site where we ultimately decided to drill are < 1 m yr$^{-1}$. Given the expected relation between surface velocity, ice thickness, and basal velocity shown in Fig. 3, this suggests that uneroded bedrock surfaces likely extend hundreds of meters below the modern ice level.

## 5.2    Local meteorology, accumulation and ablation

As noted above, mean annual temperature in the region of the Pirrit Hills is approximately -26°C. Firn depths vary from zero

over blue-ice areas to at least ∼36 m, as measured in access holes for subglacial drilling described below. One-year ice-motion stakes placed around Harter Nunatak in 2015 and re-surveyed in 2016 (Fig. 4) showed changes in the ice surface of ± 0.4 m,



comparable to the height of sastrugi in the area. Given these values, accumulation in the vicinity of the nunatak appears to be low, suggesting that the firn and ice column overlying the targeted drill sites accumulated upstream, but nearby.

Regionally, snow-bearing winds descend the ice sheet and cross the Pirrit Hills from southwest to northeast. Snow has accumulated into an embankment upwind of the Pirrit Hills, which rises over a distance of ∼5-10 km to the level of the col between Mt. Tidd and Mt. Goodwin (Fig. 4a). The ice surface drops ∼600 m across this obstruction to the northeast, where

the massif is bordered by a 1-2 km wide blue-ice ablation zone. This geometry results from descending warm, turbulent, foehn-like winds that ablate the ice surface in the lee of the mountains (cf. Bintanja, 1999). Any changes in wind direction during collapse events could modify this pattern of accumulation and ablation, and potentially induce ice-thickness changes comparable to amounts expected from dynamical thinning (Fig. 2). Atmospheric modeling suggests that surface winds near the Pirrit Hills may vary slightly in direction and magnitude during collapse events (Scherer et al., 2016); however a fundamental

reconfiguration of accumulation and ablation areas appears unlikely.

At smaller scales around the Pirrit Hills, obstructions such as minor peaks and low bedrock ridges can reverse the spatial pattern of snow erosion and deposition, with wind scoops on the upwind side and aprons of snow and ice in the lee. The ridge shown in Fig. 5d exhibits a combination of such features. Their distribution around bedrock uncovered by small-scale deglaciation is difficult to predict, and could confuse cosmogenic nuclide records by shielding rock above, or exposing rock

below the regional ice sheet surface. When drilling to shallow bedrock these potential complications are probably best avoided by targeting the crests of subglacial ridges.

## 5.3 Selected drill sites near Harter Nunatak

The subglacial topography northeast of the Pirrit Hills appears to be that of a large cirque, its central basin flanked by subglacial ridges descending from Mts. Tidd and Turcotte and re-emerging at Harter and John Nunataks respectively (Fig. 4). Regional

ice flow crosses these ridges obliquely from west to east, producing steep, locally-crevassed slopes along much of their length, precluding drilling into the underlying bedrock. However, in the course of radar reconnaissance in 2013 we circled both outlying nunataks and identified a subglacial ridge extending northwest of Harter Ntk. Unlike the major ridges radiating from the Pirrit Hills massif, this ridge lies roughly parallel to ice flow and is overlain by a featureless firn surface dipping gently towards the nunatak. As shown in Fig. 4, the ridge is asymmetric with steeply dipping southwest and gently dipping (∼ 20°)

northeast flanks. This ridge became the chosen target for two drillholes in 2016-17. Radar methods and additional survey data are given in the Supplementary Information.

The trend of the ridgecrest (Fig. 4) is almost perpendicular to the prevailing wind direction. Combined with the steepness of the upwind face, this might be expected to lead to a lee-side ablation zone in any deglaciation that removed hundreds of meters of regional ice, mimicking the gross morphology of ice surfaces around the Pirrit Hills discussed above. However, smaller

deglaciations that only exposed tens of meters of the ridgecrest may have left a lee-side snowbank, similar to that downwind of Harter Nunatak at the present day. In siting a shallow subglacial drillhole here, we therefore aimed to drill into crest of the ridge itself.



During the 2016-2017 summer, we planned to drill at two sites above this ridge with ice thickness of 100 m and 200 m, respectively. The target for the 100 m borehole was the ridge crest at site RB-1 shown in Fig. 4b. Because we were unable to image deeper portions of the ridgecrest, we sited the second borehole northeast of the crest, above the gently-dipping ridge flank. Circulation of drilling fluid in the RB-1 borehole hydrofractured the basal ice when the hole was within 10 m of the bed, forcing the borehole to be abandoned. Because the ice flow at this site is oblique to the ridge, it is possible that the basal

ice is subject to extensional stresses, which could facilitate brittle failure. To maximize the likelihood of reaching the bed, we moved the drill to site RB-2 (Fig. 4b), which is upstream of the ridgecrest with respect to ice flow. The firn surface here dips toward the ridge, suggesting that the basal ice may be in a state of compression and more resistant to fracturing. A subglacial bedrock core was successfully recovered at this site from a depth of 150 m. If past ice levels at this site are representative of regional ice levels, the measurements on the bedrock core should provide constraints on whether Robin Subglacial Basin,

below Institute and Möller Ice Streams, deglaciated in the past. Because we were unable to recover bedrock from a shallower site, the measurements will not necessarily be able to detect deglaciations in which less than 150 m of thinning occurred.

## 6   Conclusions

Measurements of cosmogenic nuclides in bedrock retrieved from below the WAIS have the potential to establish whether and when marine-based portions of the ice sheet deglaciated in the past. The potential of this method, however, requires that drill

sites meet three basic criteria: (i) local ice levels must be drawn down significantly during collapse events; (ii) the subglacial bedrock must contain minerals in which useful cosmogenic nuclides can be measured; and (iii) the cosmogenic-nuclide record, which is primarily produced in the top few meters of exposed bedrock, must remain continuously protected from erosion. These criteria are also applicable to subglacial drilling projects testing whether marine basins in East Antarctica deglaciated in the past. Sites that are expected to be most indicative of past ice-sheet extent are located adjacent to deep marine basins, such

as Bentley Subglacial Trench, where maximum ice loss would occur during a collapse event. Because ice levels at each of the potential drill sites discussed above are sensitive to the deglaciation of different sectors of the WAIS, subglacial samples from multiple sites will ultimately be required to fully determine the configuration of the ice sheet during past interglacial periods.

The Pirrit Hills are located in the Weddell Sea sector, midway between the grounding line and the divide. Ice-sheet modeling suggests that deglaciation of Robin Subglacial Basin induces thinning of a few hundred meters at the Pirrit Hills. Field

observations and cosmogenic-nuclide measurements indicate that ice levels at this site have, indeed, been lower in the past, and multiple lines of evidence suggest that uneroded bedrock extends hundreds of meters below the modern ice surface. Ice-penetrating radar surveys revealed a gently plunging subglacial ridge extending from nearby Harter Nunatak, and, in the 2016-2017 summer, an 8 m bedrock core was extracted from the ridge from below 150 m of ice. If ice levels at the drill site are representative of the region, measurements on the core should constrain past drawdowns of grounded ice in the Weddell Sector

of the ice sheet.





*Code and data availability.* The code for the PSU-3D ice-sheet model is available on request from DP. Ice-thickness, strain, and accumulation data are available from the corresponding author.

## Appendix A: Supplementary Material

### A1   $^{10}$Be and $^{26}$Al measurements

Samples were prepared for $^{10}$Be/$^9$Be and $^{26}$Al/Al measurements at the University of Washington. For each sample, we crushed and sieved the rock at 250-500 microns, and purified quartz using surfactants and dilute HF etching (Kohl and Nishiizumi, 1992). Samples were then dissolved in HF, after which total Al concentrations were measured on aliquots of the solution via inductively coupled plasma optical emission spectrometry. Be and Al were isolated using ion-exchange chromatography (Ditchburn and Whitehead, 1994), and Be and Al isotope ratios were measured at the Lawrence Livermore National Laboratory Center for Accelerator Mass Spectrometry (LLNL-CAMS). Be isotope ratios were measured relative to the ICN 01-5-4 Be standard, assigned a $^{10}$Be/$^9$Be ratio of $2.851 \times 10^{-12}$ (Nishiizumi et al., 2007). Be-10 and Al-26 production rates by spallation are based on the global calibration dataset by Borchers et al. (2016), adjusted for altitude and latitude using the scaling scheme of Lal (1991) and the relationship between Antarctic air pressure and elevation (Stone, 2000). Production rates by muons are calculated using the method of Heisinger et al. (2002a, b).

### A2   Ice-penetrating radar surveys

We used a radar built by the Center for Remote Sensing of Ice Sheets (CReSIS) with a center frequency of 750 MHz. This radar has a cross-track antenna array consisting of two widely-spaced transmitters and eight receivers designed to identify and locate off-nadir reflections. After processing, a full tomographic reconstruction of the subglacial topography was constructed. However, prior to the 2016-17 subglacial drilling season at the Pirrit Hills, ice-thickness errors were discovered in this reconstruction, and so the bed at the drill site was re-surveyed with a Geophysical Survey Systems Inc. (GSSI) SIR-4000 control unit and a 100 MHz monostatic transceiver. These data are shown in Fig. 4b. The survey consisted of parallel lines spaced 50 m apart and oriented obliquely to the subglacial ridge, as well as other exploratory lines around Harter Nunatak. The survey was conducted by towing the antenna on foot at a pace less than 0.5 m sec$^{-1}$. The data were processed using GSSI RADAN software. Distance and elevation corrections were applied to the data. To reduce noise, the data were stacked and a bandpass filter was applied. To calibrate radar wave propagation velocities and thereby improve ice-thickness estimates, we measured firn density to a depth of 38 m and used the relationship of Kovacs et al. (1995) to estimate how the real dielectric constant varied with depth. We extrapolated the depth-density relationship to deeper depths by fitting the data using a firn-densification model (Herron and Langway, 1980). Drilling at the RB-2 site (Fig. 4b) encountered the bed at a depth of 150 m, confirming ice-thickness estimates from radar surveys of $152 \pm 10$ m.





## A3 Strain and accumulation measurements

Sixteen stakes were arranged in a grid around Harter Nunatak, and two additional stakes were placed over the subglacial ridge (Fig. 4b). Stakes were installed and surveyed in December 2015, and measurements were repeated in December 2016. Stake positions were surveyed with a Trimble 5700 GPS receiver with a Zephyr Geodedic antenna. No base station was used for the first survey; the second survey was able to use a POLENET GPS station located on Harter Nunatak as a base station. Mean

5  velocity uncertainty is 0.23 m yr$^{-1}$.

   Change in snow surface height was measured at each stake. Over the one-year period, height change varied from -39 cm yr$^{-1}$ to +43 cm yr$^{-1}$, with a mean of -5 cm yr$^{-1}$. There is no apparent spatial pattern to the results, which are comparable to the amplitude of sastrugi in the area.

*Competing interests.* The authors declare that they have no conflict of interest.

10  *Acknowledgements.* Support for this work was provided through US National Science Foundation (NSF) grants 1142162 and 1341728, and the United States Antarctic Program. P.S. received funding from the NSF Graduate Research Fellowship Program. We thank Maurice Conway and Paul Koubek for assistance in the field, Seth Campbell for guidance in collection and processing of radar data, Taryn Black and Mika Usher for field and laboratory assistance. Geospatial support for this work provided by the Polar Geospatial Center under NSF OPP awards 1043681 and 1559691.





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





**Figure 1.** Maps of West Antarctica, colored for bedrock elevation (Fretwell et al., 2013) and the velocity of grounded ice (Rignot et al., 2011). White circles show the locations of possible drill sites discussed in the text. Black circles show the locations of other sites discussed in the text.





**Figure 2.** Results of a 5 Myr ice-sheet simulation. **(a)** variations in the mass of the WAIS, where the WAIS is taken to be the portion of the ice sheet between W 180° and W 50°. The maps show the WAIS at its smallest **(b)** and largest **(c)** extents of the past 1 Myr. These occur at 205 and 625 kyr B.P., respectively. The modern grounding line is shown for comparison. The maps are colored to show how much thinner or thicker the ice sheet was at these times relative to present. **(e-s)** the relationship between WAIS mass and local ice thickness at candidate drill sites. Each point represents a single 5000 year model timestep and is colored to distinguish ice sheet-behavior during the late Pleistocene (blue: 0.8 Myr BP - present) from the early Pleistocene (yellow: 2.58-0.8 Myr BP) and Pliocene (red: 5.0-2.58 Myr BP). Horizontal dashed lines on some plots represent complete deglaciation of the site. Note that although the vertical axes are limited to values between -1000 to 1000 m, during large deglaciations Mt Resnik and Pagano Nunatak become completely ice free, and the local ice thins up to ~2600 m and ~1700 m, respectively.





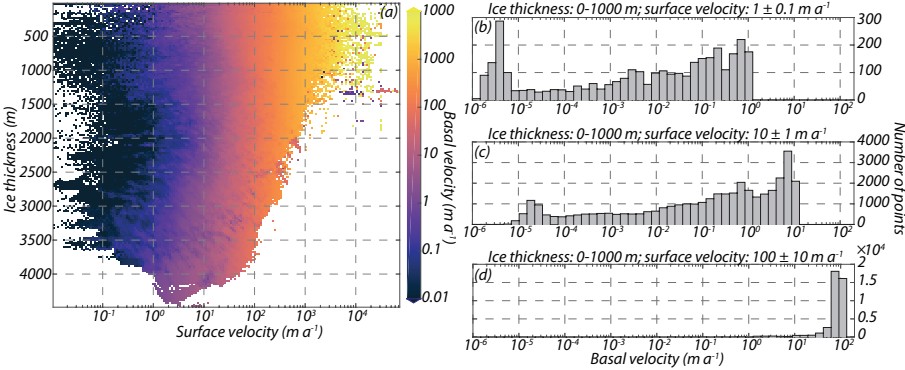

**Figure 3. (a)** The relationship between surface velocity, ice thickness, and basal velocity, as predicted by the 5 Myr ice-sheet model. Results for the full ice sheet are binned by surface velocity and ice thickness, and each bin is colored by its average basal velocity. This averaging hides considerable variability in the actual range of basal velocities within each bin. Therefore, in panels **(b-d)**, we provide histograms of basal velocity for points in the model where the ice is less than 1000 m thick, for surface velocities of $\sim$1, $\sim$10, and $\sim$100 m/yr. At depths relevant to subglacial drilling, significant subglacial erosion is unlikely in areas where the surface velocity is less than $\sim$10 m/yr (see Fig. 1).

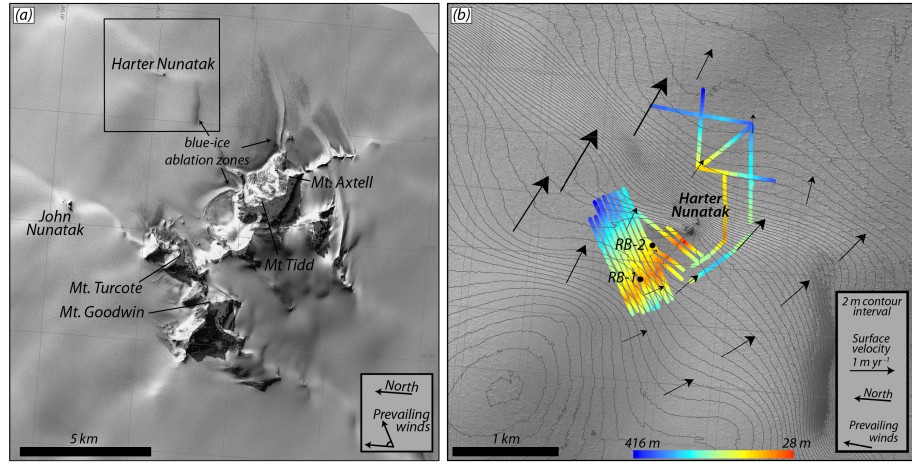

**Figure 4. (a)** WorldView satellite imagery (copyright DigitalGlobe, Inc.) of the Pirrit Hills. Black box shows the location of panel **(b)**. **(b)** Map of Harter Nunatak (center) and the surrounding area. Elevation contours are derived from WorldView satellite imagery (DEM created by the Polar Geospatial Center). Colors represent ice thickness as measured with radar surveys. Arrows represent ice velocity and were measured by repeat stake surveys. The ice velocity measured closest to the RB-2 drill site is $\sim$0.25 m yr$^{-1}$. Changes in ice-surface elevation were also measured at these stakes.



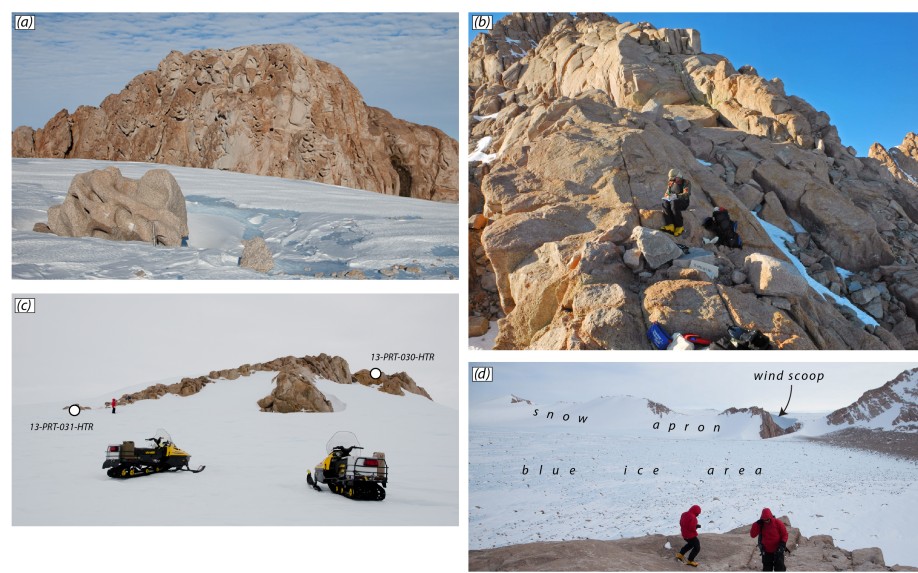

**Figure 5. (a)** Bedrock ridge located southeast of Mt. Axtell at the Pirrit Hills (Fig. 4). The granite is oxidized and displays large cavernous weathering pits (ice axe in foreground for scale). Such weathering features are common at the Pirrit Hills, both on higher mountain flanks and intersecting the modern ice surface as shown here. **(b)** Photo looking up the NE ridge of Mt Axtell. Glacially-deposited boulders rest on the more oxidized bedrock of the ridge. The depositional limit is ∼15 m above the boulders in the foreground. **(c)** Photo of Harter Nunatak, showing the location of bedrock samples collected for cosmogenic-nuclide measurements. **(d)** Back side of the bedrock ridge shown in panel A. The ridge is orthogonal to the surface winds. Ice levels on the upwind side of the ridge are ∼90-130 m higher than at the blue-ice ablation zone in the lee. The ridge is directly flanked by a wind scoop on its upwind side (not visible in panel **(a)**) and a snow apron on its downwind side.





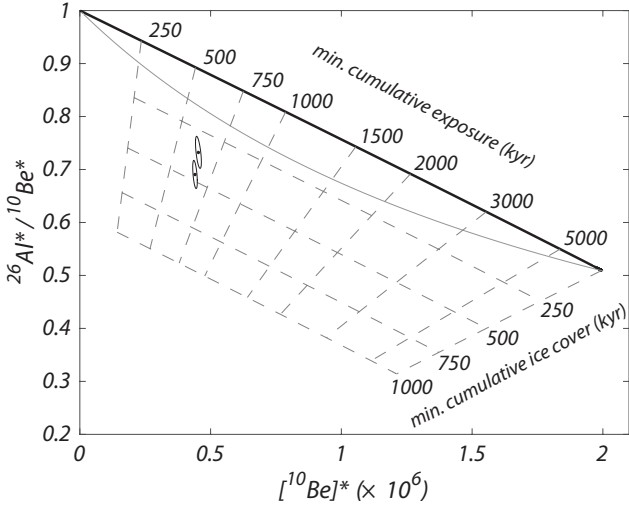

**Figure 6.** $^{10}$Be and $^{26}$Al data for the bedrock samples from Harter Nunatak shown in Fig. 5c. Nuclide concentrations are normalized to the surface production rate for each sample ($N^* = N/P$, where $N$ is the $^{10}$Be or $^{26}$Al concentration and $P$ is the local production rate of that nuclide). Ellipses represent 1 $\sigma$ uncertainty regions. Continuously exposed and uneroded surfaces will plot along the black line near the top. Continuously exposed and eroding surfaces will plot between the black line and the solid gray line. Samples plotting below the solid gray line, such as those from Harter Nunatak, require at least one episode of ice cover following prior exposure. Dashed contours show lower limits on the cumulative exposure and cumulative ice cover experienced by the samples.

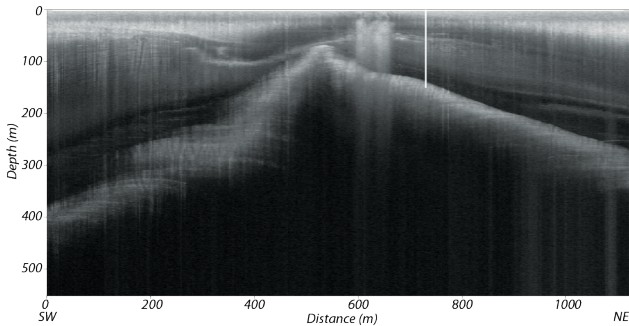

**Figure 7.** Unmigrated radar profile collected with the CReSIS accumulation radar showing the location of the 150 m RB-2 borehole to the bed. Profile is oriented perpendicular to the ridge.



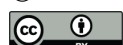

| Sample | Latitude | Longitude | Elevation (m) | Thickness (cm) | Density (g cm$^{-3}$) | Horizon correction | [Be-10] (10$^5$ atoms g$^{-1}$) | [Al-26] (10$^5$ atoms g$^{-1}$) |
|---|---|---|---|---|---|---|---|---|
| 13-NTK-030-HTR | -81.10278 | -85.14707 | 1296.8 | 3.25 | 2.53 | 0.999 | 75.2 ± 1.3 | 384.9 ± 8.8 |
| 13-NTK-031-HTR | -81.10242 | -85.14913 | 1290.1 | 1.75 | 2.58 | 0.998 | 73.6 ± 0.97 | 354.9 ± 7.8 |

**Table A1.** Sample information and cosmogenic-nuclide concentrations. Errors (±1σ) include laboratory procedural uncertainties and individual AMS measurement errors.