# Peer review of "West Antarctic sites for subglacial drilling to test for past ice-sheet collapse"

_The Cryosphere, 2018_

## Referee Comment (RC1) · Anonymous Referee #1 · 22 Jun 2018

General Comments In this paper, the authors outline the considerations required for successful cosmogenic nuclide sampling of the bed in order to test ideas for reduced past ice sheet extent. The authors outline and then apply a number of criteria to analyse the areas which would be most likely to record restricted ice extent histories. Criteria for successful sampling include an understanding of lithology which relies on knowledge of exposed outcrops and an estimation of whether sites have undergone subglacial erosion. Resulting potential sites are assessed via the use of numerical models of glacial-interglacial behaviour to determine those most likely to sensitively record the degree, timing and frequency of different minimal interglacial ice configurations whilst also being acceptable under the lithological and erosion criteria. Following these analyses, the authors report on a site survey driven by the above criteria and the

successful drilling to bedrock and extraction of a subglacial rock sample suitable for cosmogenic isotope analysis. This is a novel paper in that it applies the whole work-flow from initial assessment to successful sampling. The paper is likely to be used as a framework for future approaches to sample planning, and will be relevant not only to those intending to do exposure age dating, but to those who seek to gain rock samples for a wider set of purposes. The paper is very clearly written. Initially when I read it I felt that the structure meant it was a paper of two halves, and I did wonder whether the site criteria and analysis should have been separate from the actual fieldwork. However, on re-reading I think that both should stay together because the field campaign does provide a test of the criteria. Overall I have a few relatively minor comments that I hope will be of use:

Specific Comments P13, line8: you say a core is successfully recovered but you should also mention the drill that was actually used to do this. You don't mention that on p2 line32 either. Between these two points in the paper you should at least mention the specific system although full details are not necessary. On page 3 lie 21 you also mention the potential that exists in extracting rock cores that extent several meters in length. I think it will be useful where you mention your drill to therefore indicate how long a core it may potentially be able to get. P5 line 18 you mention an overarching criteria that drill sites should be located in sites of largest thickness change. This criteria is in tension with the notion of looking for sites with minimal erosion because thickness is a key control on whether the base reaches pressure melting point (and this erodes). Therefore, you could consider this as a discussion point when you are discussing either erosion/preservation or when discussing thickness change. Page 6 line 15: Airbourne geophysical data was not mentioned as a potentially useful tool in the initial steps of analysis. Obviously it is not a replacement for more localised site survey, but could the use of RES flight lines be an additionally useful input to the analysis? Could it add to the context provided by bedmap2? Page 9 line 15. Does the technicue by Ross et al (2013, GSAB) help in any way for site selection? Ross uses MODIS and RADARSAT data to highlight where subglacial ridges lie. P12 lines 27-32: Could variability around stoss

and lee snow build up or scour cause complications when interprating exposure history in any rocks from just beneath the ice in such locations? Or are depth criteria relating to shielding also accounting for potential variability in scour and build up? P13 line 10: You might provide an indication of the length of rock core you recovered. In addition, can you say anything else about the actual subglacial conditions encountered? Is there any evidence, from the top surface of the rock, or erosion or weathering? In other words, were your criteria relating to ensuring no erosion etc. actually supported by the physical characteristics of the sample? As a final point, could you discuss anything that you would do differently following the field sampling. What did you learn that might alter any criteria in your initial analysis or site selection?

Technical Corrections: Fig 1: The variation in the orange colors in the lower panel does not really communicate clearly the variability in elevation. Can an alternative color scheme be used to illustrate topography more clearly? Fig 2: I would find it useful to also see a map of a modelled intermediate retreat as opposed to just seeing the most significant retreat. It would help illustrate the point that you select a site that is sensitive to the different scales of interglacial minima. In the lower panels of mass relative to present, you don't indicate what the white circles mean. Finally, in this figure/caption, it would be good to be clear about why you chose those particular breaks in time for the different colours in mass loss plots. Under this categorisation, and given you make the point in the text that during some glacials there is thinning and in some interglacials there is thickening, I wondered whether categorising in these 3 time bands was as useful as it could be. This is because I can't tell whether, in the Late Pleistocene for example, the thickening relates to interglacial or glacial parts of the time period. Instead, I wonder whether it could be more useful to show 3 different colours which relate to mass under either 'glacial', 'interglacial' or 'superinterglacial' conditions. This would provide a way to more easily compare site conditions under particular scales of retreat and would also allow us to see where both thinning or thickening during interglacials occurs (given you mention that growth occurs in some regions in interglacial times).

---

## Referee Comment (RC2) · N. A. Lifton (Referee) · 8 Jul 2018

**GENERAL COMMENTS**

This manuscript presents criteria for evaluating sites for subglacial drilling to evaluate significant past ice sheet thinning beyond current conditions, specifically in West Antarctica but which would also applicable to other locations. This is an emerging field as new drilling technologies and capabilities appear for remote, logistically challenging, and environmentally sensitive applications in Antarctica and other ice-covered regions. Spector et al. do an extremely thorough job in this respect, in my opinion, beginning with description and analysis of ice-sheet modeling results and moving on to detailed site-specific criteria for consideration. In my view this is an impressive, well-written

manuscript – it is extremely clear and concise but with considerable detail and similarly clear figures, providing a very useful framework for those in the community considering similar projects. I only had a few minor comments, below, but easily recommend acceptance with minor revision. I'm pleased to say that is one of the best manuscripts I've read in a while.

Nat Lifton

SPECIFIC COMMENTS

Pg 12, Ln 15: Concerning the last sentence, if the ridge is oriented subparallel to the wind direction then even the ridge crests might also be affected by wind scoops and snow aprons, so it would be safest to avoid ridges in that orientation.

Pg 12, Ln 24: Figure 7 is a clearer demonstration of the asymmetry in my opinion

Pg 13, Ln 8: How long is the core?

Pg 14, Ln 10: I would argue that that the LSDn scaling model best explains global production rates overall, and should be used instead of Lal (okay to present Lal also, though). Also, the ERA-40 reanalysis gives very similar pressure results to those in Stone (2000). Muon production should be modeled following Balco (2017) Production rate calculations for cosmic-ray-muon-produced 10Be and 26Al benchmarked against geological calibration data. Quaternary Geochronology 39, 150–173. doi:10.1016/j.quageo.2017.02.001

Fig 4: A range of prevailing wind directions is shown in 4a, but only a single direction in 4b - seems like it should be a range as well.

Figure 6: This plot shows the ratios of (N*lambda)/P (or equivalently, N/(P*tau)), not just N/P as stated in the caption (N/P would look similar to the more typical curved two-isotope plot but with the 26/10 axis scaled from 0-1). The text in the caption should be changed to correct this.

[Figure]

---

## Author Comment (AC1) · 30 Jul 2018

We thank Nathaniel Lifton for his constructive comments that have improved the manuscript. Comments from the reviewer (bold) and our responses follow. We will upload the new version of the manuscript and supplement separately.

**GENERAL COMMENTS**

**This manuscript presents criteria for evaluating sites for subglacial drilling to evaluate significant past ice sheet thinning beyond current conditions, specifically in West Antarctica but which would also applicable to other locations. This is an emerging field as new drilling technologies and capabilities appear**

**for remote, logistically challenging, and environmentally sensitive applications in Antarctica and other ice-covered regions. Spector et al. do an extremely thorough job in this respect, in my opinion, beginning with description and analysis of ice-sheet modeling results and moving on to detailed site-specific criteria for consideration. In my view this is an impressive, well-written manuscript - it is extremely clear and concise but with considerable detail and similarly clear figures, providing a very useful framework for those in the community considering similar projects. I only had a few minor comments, below, but easily recommend acceptance with minor revision. I'm pleased to say that is one of the best manuscripts I've read in a while.**

**Nat Lifton**

We thank Nathaniel Lifton for his support of the manucript, and we have addressed his specific comments as described below.

**SPECIFIC COMMENTS**

**Pg 12, Ln 15: Concerning the last sentence, if the ridge is oriented subparallel to the wind direction then even the ridge crests might also be affected by wind scoops and snow aprons, so it would be safest to avoid ridges in that orientation.**

In this paragraph we have attempted to provide drill-site considerations that are generally applicable to alpine landscapes found in West Antarctica. As mentioned in the second paragraph of Section 5.3, an important point is that wind scoops and snow aprons are only a potential problem for deglaciations in which the ice surface is brought within a few tens of meters of a ridge crest, which is similar to the typical dimensions

of wind scoops and snow aprons. These issues will probably be irrelevant for most deglaciations in which the ice surface is more than a few tens of meters above or below a ridgecrest. We maintain that even in the case of a ridge oriented parallel to the prevailing winds, the crest will have greater immunity to these problems than the flanks.

**Pg 12, Ln 24: Figure 7 is a clearer demonstration of the asymmetry in my opinion**

Updated to "As shown in Figure 7..."

**Pg 13, Ln 8: How long is the core?**

Updated to: "An 8 m subglacial bedrock core..."

**Pg 14, Ln 10: I would argue that that the LSDn scaling model best explains global production rates overall, and should be used instead of Lal (okay to present Lal also, though). Also, the ERA-40 reanalysis gives very similar pressure results to those in Stone (2000). Muon production should be modeled following Balco (2017) Production rate calculations for cosmic-ray-muon-produced 10Be and 26Al benchmarked against geological calibration data. Quaternary Geochronology 39, 150–173. doi:10.1016/j.quageo.2017.02.001**

We have switched our production-rate calculations to use LSDn scaling and the method of Balco (2017) for muon production. We are still using the relationship between elevation and Antarctic air pressure of Stone (2000).

**Fig 4: A range of prevailing wind directions is shown in 4a, but only a single direction in 4b - seems like it should be a range as well.**

We thank the reviewer for this careful observation. The difference between the wind-direction indicators in the two figures is actually purposeful. In this region of Antarctica, the winds generally flow from the southwest. The range shown in Figure 4a reflects the fact that the winds are forced to flow around the Pirrit Hills massif, and so the orientation of the prevailing winds is not uniform in the region, but varies with location around the mountains. At a given location, the winds are relatively constant in direction; hence for Figure 4b we used a single vector that is representative of the wind direction in the domain of the map. We have updated the figure caption to clarify this.

**Figure 6: This plot shows the ratios of (N*lambda)/P (or equivalently, N/(P*tau)), not just N/P as stated in the caption (N/P would look similar to the more typical curved two-isotope plot but with the 26/10 axis scaled from 0-1). The text in the caption should be changed to correct this.**

We believe that this comment is incorrect. The concentrations shown in this figure are, in fact, normalized to surface production rates (N/P). We do not use N*lambda/P, as indicated by the reviewer. The 26/10 axis is scaled from 0-1 (in the figure it is shown from 0.2-1 to focus on the area of interest). The X-axis is linear, which causes the simple exposure line and the contours of exposure and ice cover to be straight lines. Two-nuclide diagrams in other publications are sometimes presented with the X-axis on a log scale, which causes these lines to be curved. This is likely the root of the confusion.

---

## Author Comment (AC2) · 30 Jul 2018

We thank the reviewer for a thorough review that have improved the quality of the manuscript. Comments from the reviewer (bold) and our responses follow. We will upload the new version of the manuscript and supplement separately.

**General Comments**

**In this paper, the authors outline the considerations required for successful cosmogenic nuclide sampling of the bed in order to test ideas for reduced past ice sheet extent. The authors outline and then apply a number of criteria to analyse the areas which would be most likely to record restricted ice extent histories. Cri-**

**teria for successful sampling include an understanding of lithology which relies on knowledge of exposed outcrops and an estimation of whether sites have undergone subglacial erosion. Resulting potential sites are assessed via the use of numerical models of glacial-interglacial behaviour to determine those most likely to sensitively record the degree, timing and frequency of different minimal interglacial ice configurations whilst also being acceptable under the lithological and erosion criteria. Following these analyses, the authors report on a site survey driven by the above criteria and the successful drilling to bedrock and extraction of a subglacial rock sample suitable for cosmogenic isotope analysis. This is a novel paper in that it applies the whole work-flow from initial assessment to successful sampling. The paper is likely to be used as a framework for future approaches to sample planning, and will be relevant not only to those intending to do exposure age dating, but to those who seek to gain rock samples for a wider set of purposes. The paper is very clearly written. Initially when I read it I felt that the structure meant it was a paper of two halves, and I did wonder whether the site criteria and analysis should have been separate from the actual fieldwork. However, on re-reading I think that both should stay together because the field campaign does provide a test of the criteria. Overall I have a few relatively minor comments that I hope will be of use:**

We thank the reviewer for their support of the manuscript, and we address their specific comments as described below.

**Specific Comments**

**P13, line8: you say a core is successfully recovered but you should also mention the drill that was actually used to do this. You don't mention that on p2 line32 either. Between these two points in the paper you should at least mention the specific system although full details are not necessary. On page 3 lie 21 you**

**also mention the potential that exists in extracting rock cores that extent several meters in length. I think it will be useful where you mention your drill to therefore indicate how long a core it may potentially be able to get.**

We agree that there was insufficient information about the drill we used. We have added text describing the drill to the final paragraph of Section 5.3: "We used the Agile Sub-Ice Geological (ASIG) Drill, a modified wireline mineral exploration drill, which is similar in design to the much larger Rapid Access Ice Drill (RAID) (Goodge and Severinghaus, 2016). The ASIG Drill is designed to be able to drill 15 m of rock core beneath 700 m of ice."

**P5 line 18 you mention an overarching criteria that drill sites should be located in sites of largest thickness change. This criteria is in tension with the notion of looking for sites with minimal erosion because thickness is a key control on whether the base reaches pressure melting point (and this erodes). Therefore, you could consider this as a discussion point when you are discussing either erosion/preservation or when discussing thickness change.**

To address this comment, we have modified the sentence to read: "Therefore, an overarching criterion is that drill sites be located in areas that experience the largest change in ice thickness during collapse events, bearing in mind the need to avoid sites where ice flow may have been erosive in the past."

**Page 6 line 15: Airbourne geophysical data was not mentioned as a potentially useful tool in the initial steps of analysis. Obviously it is not a replacement for more localised site survey, but could the use of RES flight lines be an additionally useful input to the analysis? Could it add to the context provided by bedmap2?**

[Figure]

We agree that airborne geophysical data can provide important information about sub-glacial topography and rock types. With regard to our field sites, the Pirrit Hills, Nash Hills, and Whitmore Mountains, airborne geophysical data were not discussed in detail because there was a paucity of such data in the areas where we were considering drilling. We do consider airborne geophysical data where they are relevant to candidate drilling sites. For example, in the last sentence of Section 2.2, we discuss the composition, form, relief, and depth below the surface of subglacial Mt. Resnik, which is based on airborne geophysical surveys.

Airborne geophysical surveys are likely to be critical in the future, where drilling targets have no surface outcrop (e.g. Mt. Resnik). In such cases, gravity and magnetic surveys would be the only way to asses lithology ahead of drilling. We did not develop this discussion in the paper; however, because we think that blind subglacial bedrock drilling for cosmogenic-nuclide measurements is too risky at this stage in the evolution of the science.

**Page 9 line 15. Does the technicue by Ross et al (2013, GSAB) help in any way for site selection? Ross uses MODIS and RADARSAT data to highlight where subglacial ridges lie.**

Ross et al. use MODIS imagery, which is influenced by the ice-surface elevation, to derive information about subglacial topography. The reviewer is correct that surface elevation, along with other surface features (e.g. roughness, crevasses) contain important information about the bed that is useful for drill-site selection. In our work around the Pirrit Hills, we did use such information, although in a less formal manner. For example, in Figure 4a, the bright areas on the ice-sheet surface generally correspond to steeper slopes in the lee of subglacial ridges. This is evident from the fact that these areas adjoin outcropping ridges which descend below the ice, and we have further confirmed this with radar surveys over the ridges between between Harter and John

Nunataks and the main massif of the Pirrit Hills. We have added a sentence to the caption of Figure 4 describing the relationship between the bright areas seen in Figure 4a and subglacial ridges.

**P12 lines 27-32: Could variability around stoss and lee snow build up or scour cause complications when interprating exposure history in any rocks from just beneath the ice in such locations? Or are depth criteria relating to shielding also accounting for potential variability in scour and build up?**

Yes, as discussed in the last paragraph of Section 5.2, features such as wind scoops or snow aprons could complicate the interpretation of past exposure and ice-cover if the ice surface is brought to within a short distance (a few tens of meters, similar to the typical dimensions of such features) above or below a bedrock drill target. As mentioned in the text, this is one of the primary arguments for targeting ridge crests, which are most likely to remain immune to these complications.

**P13 line 10: You might provide an indication of the length of rock core you re-covered. In addition, can you say anything else about the actual subglacial con-ditions encountered? Is there any evidence, from the top surface of the rock, or erosion or weathering? In other words, were your criteria relating to ensuring no erosion etc. actually supported by the physical characteristics of the sample?**

We have added the following text to the final paragraph of Section 5.3: "An 8 m sub-glacial bedrock core was successfully recovered at this site from a depth of 150 m. The core surface shows no smoothing or striations indicative of subglacial abrasion, consistent with cold-based ice cover. Nor is there evidence of oxidation or other sub-aerial weathering features. Details of the core and analyses on it will be described in forthcoming publications."

[Figure]

**As a final point, could you discuss anything that you would do differently following the field sampling. What did you learn that might alter any criteria in your initial analysis or site selection?**

We remain satisfied with our decision to drill at the Pirrit Hills, as well as the subglacial ridge located near Harter Nunatak. As mentioned in the manuscript, we would have preferred to target the crest rather than the flank of the ridge, but we were unable to image deeper portions of the ridgecrest. This emphasizes the need to do careful and thorough radar reconaissance prior to drilling, and we believe that this point is clearly made in the manuscript.

The hydrofracture that occurred during our first drilling attempt meant that we recovered only one subglacial rock core rather than two, as we had hoped. Obtaining multiple cores from multiple depths from multiple sites within the ice sheet will ultimately be necessary to establish the configuration of the ice sheet during interglacial periods. Therefore, understanding the fragility of basal ice in areas where the ice sheet is thin, the basal ice is cold, and the ice flows over complex alpine topography will be important for future subglacial drilling projects. Much of the ice core recovered from the Pirrit Hills was 'milky' in appearance due to internal fractures. Learning whether these fractures resulted from strain at the base of the ice column or whether they were induced by the drilling process will be important so that either (i) future drill sites can be selected in areas where damaged basal ice is less likely, or (ii) drilling methods can be modified so as to reduce stresses on the ice.

**Technical Corrections:**

**Fig 1: The variation in the orange colors in the lower panel does not really communicate clearly the variability in elevation. Can an alternative color scheme be used to illustrate topography more clearly?**

The primary purpose of the color scheme is to identify where the bed is below sea level and where it is above. Communicating exact elevations is much less important. In order to keep the map visually simple, we purposefully chose a low-contrast color scheme for the areas above sea level. To clarify elevation above sea level we have made the dark oranges darker and the light oranges lighter. We have also updated the caption to clarify the purpose of the figure.

**Fig 2:**

**I would find it useful to also see a map of a modelled intermediate retreat as opposed to just seeing the most significant retreat. It would help illustrate the point that you select a site that is sensitive to the different scales of interglacial minima.**

We agree that drill sites should be located in areas where the ice thickness is sensitive to WAIS deglaciations of different magnitudes. This argues, for example, for drilling at sites such as Mt. Resnik, where there is a large and continuous thinning as the WAIS diminishes (see Fig. 2r). For simplicity, we did not include a map showing an intermediate retreat. Information about how sites respond to different magnitudes of WAIS deglaciation is communicated in figure panels d-s (see also our response to the comment below).

**In the lower panels of mass relative to present, you don't indicate what the white circles mean.**

Updated the caption to include the sentence: "White circles represent the conditions at 205 and 625 kyr B.P., which correspond to the maps in panels (b) and (c), respectively."

**Finally, in this figure/caption, it would be good to be clear about why you chose those particular breaks in time for the different colours in mass loss plots. Under this categorisation, and given you make the point in the text that during some glacials there is thinning and in some interglacials there is thickening, I wondered whether categorising in these 3 time bands was as useful as it could be. This is because I can't tell whether, in the Late Pleistocene for example, the thickening relates to interglacial or glacial parts of the time period. Instead, I wonder whether it could be more useful to show 3 different colours which relate to mass under either 'glacial', 'interglacial' or 'superinterglacial' conditions. This would provide a way to more easily compare site conditions under particular scales of retreat and would also allow us to see where both thinning or thickening during interglacials occurs (given you mention that growth occurs in some regions in interglacial times).**

We chose to color the points by time period (Pliocene, early Pleistocene, and late Pleistocene) because the modeled ice sheet exhibits significantly different behavior during these three periods. The Pliocene WAIS is small and exists only on areas of high topography. The Pleistocene WAIS is considerably larger and oscillates between expanded and collapsed configurations on ∼40 kyr timescales during the early Pleistocene and ∼100 kyr timescales during the late Pleistocene. These contrasts in behavior are shown for the WAIS in panel (a) at the top of the figure.

For individual sites, the distinction between 'glacial', 'interglacial', and 'super-interglacial' behavior is contained in the X-axis of these site-by-site plots. Points on the left of each plot predict the sites response to 'super-interglacials' when the WAIS was greatly reduced; points on the right show the site's response through glacial periods.

---

## Author Comment (AC3) · 30 Jul 2018

Dear Dr. Stroeven (editor),

We thank the reviewers for their helpful feedback, which has improved our manuscript. In the Interactive Discussion, we have replied to each Reviewer directly. Attached to this comment is a pdf of our revised manuscript.

We would also like to bring to your attention the fact that we have made some changes to the manuscript regarding issues that were not raised by the Reviewers. These changes are described below.

[Figure]

In the last paragraph of Section 5.3, we have changed the sentence which read "Circulation of drilling fluid in the RB-1 borehole hydrofractured the basal ice...". The new sentence reads "An unexplained hydrofracture of the basal ice of the RB-1 borehole...".

After our initial manuscript submission, we realized that the present-day grounding line in our ice-sheet simulation was located upstream of Robin Subglacial Basin in the Weddell Sea sector of the WAIS (compare to Fig. 1). This misfit affects the plots in Fig. 2e-s because, at sites upstream of this area, thinning during interglacial periods is underestimated and thickening during glacial periods is overestimated. To address this, we have taken the following actions.

(i) We have added a figure (Fig. 3 in the revised manuscript) showing the misfit between the modeled and the observed present-day ice sheet. This figure shows that the model does reasonably well in most areas, but very poorly in the region of Robin Subglacial Basin.

(ii) We added a paragraph to the end of Section 3 explaining this misfit and its consequences.

(iii) We have added text to the caption of Fig. 2 to make the reader aware of this issue. In the pdf attached to this comment, the caption gets cut off by the page break. The text that we have added to the caption reads: "As shown in Fig. 3 and as discussed in the text, the modeled present-day ice sheet places the grounding line upstream of Robin Subglacial Basin (compare to Fig. 1). A result of this misfit is that, for sites upstream of this area, ice-thickness changes shown in panels (e-s) underestimate the thinning during interglacial periods and overestimate the thickening during glacial periods."

We hope that you find our edits satisfactory for publication of the manuscript in The Cryosphere.

Kind regards,

Perry Spector (on behalf of all co-authors)

Please also note the supplement to this comment:
https://www.the-cryosphere-discuss.net/tc-2018-88/tc-2018-88-AC3-supplement.pdf

[Figure]

**Supplement:**

[revised manuscript text omitted]